# Fever Detection with Infrared Thermography: Enhancing Accuracy through Machine Learning Techniques

Parsa Razmara*
*Biomedical Engineering Department*
*University of Southern California*
Los Angeles, USA
prazmara@usc.edu

Tina Khezresmaeilzadeh*
*Electrical Engineering Department*
*University of Southern California*
Los Angeles, USA
khezresm@usc.edu

B. Keith Jenkins
*Electrical Engineering Department*
*University of Southern California*
Los Angeles, USA
bjenkins@usc.edu

*Abstract*—The COVID-19 pandemic has underscored the necessity for advanced diagnostic tools in global health systems. Infrared Thermography (IRT) has proven to be a crucial non-contact method for measuring body temperature, vital for identifying febrile conditions associated with infectious diseases like COVID-19. Traditional non-contact infrared thermometers (NCITs) often exhibit significant variability in readings. To address this, we integrated machine learning algorithms with IRT to enhance the accuracy and reliability of temperature measurements. Our study systematically evaluated various regression models using heuristic feature engineering techniques, focusing on features' physiological relevance and statistical significance. The Convolutional Neural Network (CNN) model, utilizing these techniques, achieved the lowest RMSE of 0.2223, demonstrating superior performance compared to results reported in previous literature. Among non-neural network models, the Binning method achieved the best performance with an RMSE of 0.2296. Our findings highlight the potential of combining advanced feature engineering with machine learning to improve diagnostic tools' effectiveness, with implications extending to other non-contact or remote sensing biomedical applications. This paper offers a comprehensive analysis of these methodologies, providing a foundation for future research in the field of non-invasive medical diagnostics.

*Index Terms*—COVID-19, Infrared Sensors, Temperature Measurement, Infrared Thermography, Machine Learning, Regression analysis, Deep Learning

## I. INTRODUCTION

As the COVID-19 pandemic continues to challenge global health systems, the adoption of advanced diagnostic tools has become crucial.

Infrared Thermography (IRT), a pre-existing technology recognized for its utility in medical diagnosis and disease monitoring, has gained renewed attention due to its non-contact and efficient method for measuring body temperature [1], [2]. This is vital for identifying elevated body temperatures, a primary indicator of infectious diseases such as COVID-19, which has a prevalence rate of fever in about 78% of confirmed adult cases. Traditional non-contact infrared thermometers (NCITs), however, exhibit significant variability in

readings, with error rates as high as 2°C [3], [4]. Consequently, the use of more advanced methods like IRT combined with machine learning algorithms becomes essential to improve the accuracy and reliability of temperature measurements, building on approaches used for COVID-19 prognosis with biomarkers and demographic information [5]. This integration not only contributes to safer and more effective public health screening practices but also aligns with the global need for improved diagnostic tools during pandemics [6], [7].

The relevance of IRT in medical diagnostics and epidemic prevention is supported by its increasing utilization in various clinical and public settings. Recent studies have highlighted the potential of IRT systems, particularly when calibrated with precise regression techniques, to provide accurate temperature readings essential for detecting potential cases of COVID-19. For example, [6] demonstrated that calibrated IRT systems, when compared with non-contact infrared thermometers (NCITs), could achieve higher clinical accuracy and repeatability in temperature measurement. Another research [3] focused on using machine learning to predict core body temperatures from IR-measured facial features, revealing that specific regions such as the temple and nose could serve as reliable indicators of body temperature.

Despite these developments, there remains a noticeable gap in the enhancement of feature engineering techniques to improve the predictive performance of these temperature measurement models. This paper introduces novel heuristic feature engineering methods that significantly enhance model accuracy, addressing shortcomings of previous studies.

The structure of the paper is as follows: Section II details our approach to data collection, feature selection, and the application of regression models, emphasizing the development of our heuristic feature engineering techniques. In Sections III and IV, we present a detailed analysis of the models' performance, offering insights into their clinical and practical implications. The paper concludes with a summary of our findings and an outlook on future research directions in Section V, exploring how these methods can be adapted to other infectious diseases and broader clinical applications.

*These authors contributed equally to this work. Responsible Authors Contacts: prazmara@usc.edu, khezresm@usc.edu

## II. METHODOLOGY

### A. Dataset

The Infrared Thermography Temperature Dataset, sourced from Forward Looking Infrared (FLIR) data, contains temperatures read from various locations of infrared images about patients, along with oral temperatures measured for each individual. The dataset comprises 1020 data points and 33 features, intended for a regression task to predict oral temperature using environmental information and thermal image readings. The dataset is divided into two groups based on ambient temperature: Group 1 (20.0 to 24.0°C) and Group 2 (24.0 to 29.0°C), both considered in this study [6], [8], [9].

*1) Data Allocation:* The final dataset, after removing some data points with missing values, consisted of 959 data points, was divided into training (669 data points) and testing sets (290 data points). This split ensured that the models were trained on a substantial portion of the data while retaining a set for unbiased evaluation of their performance. The target output is the oral temperature measured in monitor mode (aveOralM).

*2) Validation Set and Hyperparameter Tuning:* Validation sets, constituting 20% of the training data, were essential for tuning models and avoiding overfitting and bias across various algorithms. Nested cross-validation was utilized for hyperparameter optimization, each parameter set evaluated across five folds to determine its effectiveness, selecting diverse populations for each fold to mitigate the risk of bias.

### B. Feature Extraction

Our feature extraction methodology comprises polynomial features, replicated features, and categorical features.

*1) Incorporating New Features with Polynomial Interaction Terms:* Polynomial interaction terms were introduced in the feature engineering process to capture complex relationships between features. This technique goes beyond linear models, enabling the model to learn from higher-order interactions and nonlinear patterns present in the data.

Polynomial features were created to emphasize the combined and nonlinear effects of original features. By squaring certain features and creating interaction terms between pairs of highly predictive features, we aimed to enhance the richness of the data representation. This approach allows the model to better capture the underlying physiological dynamics, leading to improved predictive accuracy and robustness.

*2) Enhancing Model Predictions with Replicated Feature Engineering:* The second method in the feature engineering process involved replicating the most predictive features to amplify the impact of key features that exhibited the strongest correlation with the target variable.

The feature replication strategy involved creating multiple replicas of the most influential features identified. Unlike traditional approaches that might assign equal importance to all features, it was hypothesized that increasing the representation of a highly predictive feature would significantly improve model performance. The impact of this replication varies across different regression methods. For example, in kNN,

it makes a difference because the distance between two data points would put more emphasis on the replicated features. In contrast, for Linear Regression with no regularization, replicating one feature to two identical features would be equivalent to doubling the weight value of the original feature, making little difference in performance.

This hypothesis was tested by empirically determining the optimal number of replications through multiple iterations. By replicating these key features several times, a decrease in the RMSE was observed, indicating enhanced model accuracy. This strategy aligns with advanced analytical practices that prioritize data-driven insights over uniform feature treatment.

*3) Encoding and Developing New Features from Categorical Variables:* This approach focused on enhancing the representation of categorical variables in the dataset, specifically Gender, Age, and Ethnicity. The Age feature, exhibiting an ordinal nature, was encoded using an OrdinalEncoder. This transformation translated the categorical age ranges into an ordered numerical format, with the mapping as follows: {'18-20': 0, '21-25': 1, '26-30': 2, '31-40': 3, '41-50': 4, '51-60': 5, '> 60': 6, '> 70': 7}, facilitating a more nuanced interpretation by our regression models.

One-hot encoding was applied to the features Gender and Ethnicity, which are nominal data without intrinsic ordering. This strategy transformed each categorical variable into a set of binary variables, one for each category, ensuring the model treats each categorical value as a separate entity without imposing any ordinal relationship among them. The encoding strategy enhances the model's ability to utilize these categorical features effectively, improving overall predictive accuracy.

### C. Feature Selection

Feature selection is a crucial step in developing a robust predictive model, involving the identification of features that enhance model accuracy and efficiency. The methodology in this study is divided into biological and physiological analysis and technical and statistical analysis. A notable consideration was given to selecting and validating features based on their physiological relevance and statistical significance, drawing from both experimental data and related scientific literature.

*1) Biological and Physiological Analysis:* The selection of important features was guided by their physiological relevance. Specifically, the focus on temperatures from facial regions, particularly the inner canthi, was informed by their high correlation with core body temperatures. This area, being perfused by the internal carotid artery, is crucial for accurate non-contact temperature measurement. The high correlation of these temperatures with oral temperatures underscores their importance as predictors in our model.

Environmental factors, such as humidity and distance, potentially play a critical role in the accuracy of thermal imaging. Although humidity is not a direct predictor of oral temperature, it affects the thermal emissivity of the skin and the measurement dynamics, indirectly impacting temperature readings. The dataset accounted for fluctuations in humidity by including it as a feature, ensuring that the temperature

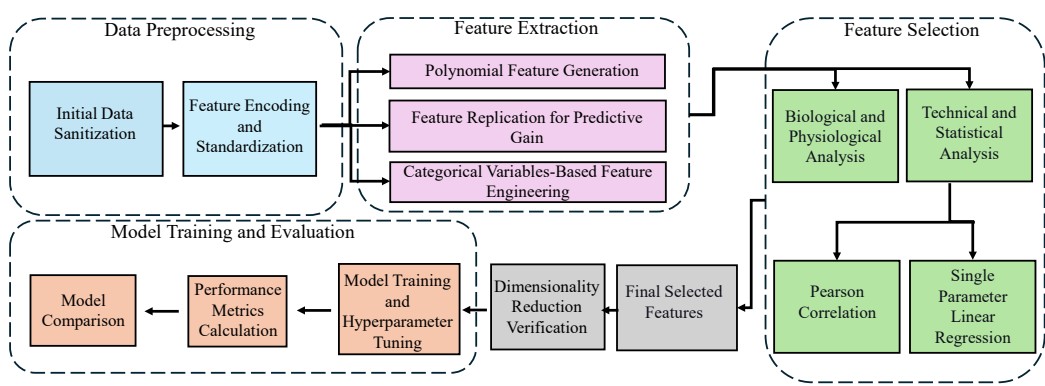

Fig. 1. Overview of the system workflow, illustrating stages from data preprocessing to model training and evaluation. The process includes initial data sanitization, feature encoding, polynomial and categorical feature engineering, biological and technical analysis for feature selection, and model training with performance evaluation.

measurements could adjust for these variations and maintain consistency and reliability under varying clinical conditions while reducing potential environmental biases.

The distance between the infrared thermography device and the subject is potentially important for ensuring accurate temperature readings. Proper distance affects the resolution and the field of view of the thermal camera, which in turn influences the accuracy of capturing temperatures at vital facial points such as the inner canthi and forehead. By including distance as a feature in the dataset, it allowed for adjustments to be made for variations, maintaining measurement accuracy.

These considerations have been integrated into the model development process, ensuring that each feature included in the final model is backed by both statistical evidence and physiological relevance. This approach enhances the predictive accuracy of the model and ensures that the model's outputs are interpretable and relevant in clinical settings. By carefully selecting features that are robustly supported by experimental data and scientific understanding, this study aims to create a reliable and effective tool for predicting oral temperatures using infrared thermography.

During the refinement process, it was observed that the distribution of data across different age groups was highly skewed, predominantly concentrated within the 21–30 age range. This imbalance raised concerns about the representativeness and effectiveness of the 'Age' feature in our models. Given the skewed distribution, the 'Age' feature's potential to contribute effectively to the predictive accuracy of the model was limited. Including age as a predictive variable could mislead the model, as it would not adequately capture the variability across the general population. Consequently, the 'Age' feature was excluded from the final model to enhance generalization and avoid biases associated with disproportionate data distribution among different age groups.

An intermediate result from the feature refinement process focused on the gender feature initially represented as a single categorical variable. To enhance the model's capacity to differentiate impacts based on gender, this variable was transformed into two distinct binary features: 'female' and 'male.' The balanced distribution of data points across these categories allowed the model to learn potential differences in oral temperature across genders without the bias that can arise from uneven sample sizes. This transformation aids in capturing any gender-specific variations in oral temperature, crucial for the precision and reliability of the predictions.

Our analysis also considered features representing temperature readings from the mouth and various points on the forehead. Supporting literature on the physiological relevance of temperature measurements at these locations suggested limited predictive power for oral temperature estimation compared to regions like the inner canthi [6], [10], [11]. Given the statistical findings and corroborating literature, features related to mouth and forehead temperatures were excluded to enhance the model's focus on more predictive variables, thereby improving its overall predictive accuracy and efficiency.

Overall $T\_Max\_1$, $canthi4Max\_1$, $canthiMax\_1$, $Max1R13\_1$, $Max1L13\_1$, $aveAllL13\_1$, $aveAllR13\_1$, $Distance$, $T\_offset$, $T\_atm$, $Humidity$, $Gender\_Female$, and $Gender\_Male$ are considered as biologically important features. Our construction of a predictive model using the Infrared Thermography Temperature Dataset involved careful selection and validation of features based on their physiological relevance. This process was informed by insights from relevant scientific literature. The integration of prior knowledge facilitated the identification of beneficial features within the physiological context of the study, enhancing the model's implementation.

*2) Technical and Statistical Analysis:* In the process of refining the set of features developed through various engineering methods, we initially used Pearson correlation coefficients to identify strong linear associations between features and the target variable, oral temperature ($aveOralM$). Recognizing that linear correlation might not capture all interactions, we complemented this with regression-based feature importance, assessing each feature's impact on model accuracy using RMSE. This dual approach ensured that both linear and non-linear relationships were effectively captured.

*a) Observations on Temperature-related Features:* The inclusion or exclusion of atmospheric temperature (T_atm) and temperature offset (T_offset) provided minimal variation in the RMSE, suggesting that these features do not significantly influence the model's accuracy.

In contrast, features related to the maximal and inner canthi temperatures, specifically those representing critical facial regions, proved to be highly significant. Their inclusion consistently lowered the RMSE, while their removal led to a noticeable degradation in model performance. This aligns with existing scientific literature that emphasizes the physiological relevance of these temperature measurements in accurately estimating core body temperatures.

Based on these findings, the feature set was refined to focus more on the Maximal and Inner Canthi temperature measurements that showed substantial predictive power and relevance. This decision was supported by both the quantitative impact on model performance and the qualitative insights derived from domain-specific studies.

*b) Mouth and Forehead Temperature Features:* Pearson correlation analysis and linear regression models were used to evaluate the relationship between each forehead and mouth temperature feature and the target variable, oral temperature ($aveOralM$). These analyses indicated that these features had low correlations and did not significantly reduce the RMSE when compared to other variables in the dataset.

Given the statistical findings and corroborating literature, it was determined that features related to the mouth and forehead temperatures did not contribute significantly to model accuracy. Consequently, these features were removed from the model, resulting in a noticeable improvement in the RMSE.

*3) Feature Dimensionality Reduction:* This involved a combination of statistical techniques to identify and enhance features with the highest predictive power.

*a) Statistical Correlation Analysis:* Initially, Pearson correlation coefficients were employed to quantify the linear relationship between each feature and the target variable, oral temperature ($aveOralM$). The absolute values of the Pearson correlation coefficients were considered to recognize both strong positive and negative correlations. This analysis pinpointed the features with the highest correlation score (absolute value of Pearson coefficient) with the oral temperature.

*b) Regression-Based Feature Importance:* To further refine feature selection, a linear regression model was employed to assess the impact of each feature on the model's accuracy. By analyzing the RMSE for models trained on single features, the features that minimize prediction errors were identified.

Table I presents the features selected based on their correlation with the target variable from (a) and their corresponding RMSE values from (b). The most effective features according to this combination of methods included $T\_Max\_1$, $canthi4Max\_1$, and $canthiMax\_1$ which exhibited the lowest RMSE values and the highest correlation scores.

This combined approach of using Pearson correlation coefficients and regression-based feature importance allowed

TABLE I
BIOLOGICALLY-RELATED FEATURES SELECTED BASED ON CORRELATION ANALYSIS AND RMSE VALUES

| Feature Name | Correlation Score | RMSE |
|---|---|---|
| $T\_Max\_1$ | 0.830394 | 0.2576921021 |
| $canthi4Max\_1$ | 0.74778 | 0.2881620275 |
| $canthiMax\_1$ | 0.74742 | 0.2878116205 |
| $Max1R13\_1$ | 0.7016 | 0.301061764 |
| $Max1L13\_1$ | 0.697585 | 0.3107211261 |
| $aveAllL13\_1$ | 0.588969 | 0.3407036569 |
| $aveAllR13\_1$ | 0.569556 | 0.3526179759 |

for a robust selection process, ensuring that the final model incorporates features with the highest predictive power.

This choice was driven by the hypothesis that the interactions between the most influential variables (as indicated by their correlation) are likely to provide significant predictive insights, especially in a biological context where many relationships are inherently nonlinear. Including only these two features in our polynomial analysis was a strategic decision to test this hypothesis without excessively complicating the model with numerous higher-order terms, which could lead to overfitting especially if extended to less influential features.

*c) Sequential Backward Selection (SBS):* In Section III-A0g, we will apply Sequential Backward Selection to systematically evaluate each feature's necessity, helping to prevent overfitting and improve generalizability.

*d) Principal Component Analysis:* After selecting the final feature set, Principal Component Analysis (PCA) was initially employed to explore further dimensionality reduction. We used the 'MLE' method for determining the number of principal components, based on [12], which optimizes system complexity by balancing bias and variance. However, applying PCA did not enhance system performance and increased the RMSE. This indicates that the feature selection process had already effectively identified the most relevant features. A comparison of RMSE results between systems using PCA and those using only feature standardization (no PCA) confirmed that excluding PCA post-feature selection provided superior accuracy. These findings underscore the robustness of the feature selection methodology, ensuring the chosen features contribute effectively to the system's performance without further dimensionality reduction. Similar methodologies in other health-related studies highlight the importance of effective feature selection and dimensionality reduction.

*4) Final Selected Features:* After this feature selection process involving Pearson correlation analysis and regression modeling, the most predictive features for oral temperature estimation were identified. The analysis, detailed in previous sections II-C1, II-C2, and II-C3, highlighted the relevance of features such as temperatures from the inner canthi, which have a high correlation with core body temperatures and are physiologically significant [6], [3]. Features with low correlation coefficients and those not significantly reducing RMSE were excluded to streamline the model, based on both statistical and physiological considerations. Consequently, the

TABLE II
FINAL ENGINEERED FEATURE SET BASED ON TECHNICAL ANALYSIS
AND SCIENTIFIC BACKGROUND KNOWLEDGE

| Final Selected Features | Selection Criteria |
|---|---|
| $T\_Max\_1$ | High corr and low RMSE, strong predictive power (+5X replications) |
| $canthi4Max\_1$ | High corr and low RMSE, strong predictive power |
| $canthiMax\_1$ | High corr and low RMSE, strong predictive power |
| $Max1R13\_1$ | Significant corr and acceptable RMSE, good predictive power |
| $Max1L13\_1$ | Significant corr and acceptable RMSE, good predictive power |
| $aveAllL13\_1$ | Moderate corr and acceptable RMSE, contributing to predictive power |
| $aveAllR13\_1$ | Moderate corr and acceptable RMSE, contributing to predictive power |
| $T\_offset$ | Important for measurement consistency; accounts for baseline temperature variations |
| $T\_atm$ | Influences thermal readings; accounting for ambient temperature variations |
| $Humidity$ | Affects skin thermal emissivity; consistent temperature measurements |
| $Gender\_Female$ | Captures gender-specific variations in temperature measurements |
| $Gender\_Male$ | Captures gender-specific variations in temperature measurements |
| $T\_Max\_1 \times canthi4Max\_1$ | Polynomial-interaction term |
| $(T\_Max\_1)^2$ | Nonlinear term |
| $(canthi4Max\_1)^2$ | Nonlinear term |

TABLE III
MODEL PARAMETERS IN REGRESSION ANALYSIS

| Model | Parameter | Values |
|---|---|---|
| 1NN Regression | n_neighbor | 1 |
| Linear Regression | l2_regularization_coef | 0.01 |
| KNN Regression | n_neighbor | {1, ..., 30} |
| Random Forest | n_estimators | {50, ..., 250} |
| SVR | C | 1 |
| | epsilon | 0.1 |
| | kernel_type | RBF |
| Quadratic Regression | max_degree | 2 |
| Weighted Linear Regression | bw_method | silverman |
| | sample_weight | 1/density |
| Binning Linear Regression | n_bins | 3 |
| CNN | activation_function | ReLU |
| | optimizer | Adam |
| | learning_rate | 0.001 |
| | epoch | 1000 |
| | batch-size | 32 |

Section II-B1, we implemented polynomial features, focusing on Feature $T\_Max\_1$ and Feature $canthi4Max\_1$. These two were selected as they are the most correlated features with Oral temperature, suggesting their strong linear relationship with the target. The resulting 3 polynomial features comprise the last group shown in Table II. In addition, five replications of $T\_Max\_1$ were integrated into the final feature set to improve its predictive influence. This decision was substantiated through extensive testing, where five replications consistently delivered superior and more reliable outcomes.

*D. Model Training and Evaluation*

We employed various regression models to address the challenges presented by Infrared data, each selected for its unique ability to capture spatial and temporal patterns.

The models range from simple linear approaches to more complex non-linear and deep learning methods. For instance, we utilized 1-Nearest Neighbor (1NN) and Linear Regression as basic benchmarks. To better handle the complexity and nuances of our dataset, more capable techniques like Support Vector Regression (SVR) with radial basis function (RBF) kernels and K-Nearest Neighbors (KNN) and Random Forest with hyperparameter optimization were also applied. Additionally, Quadratic Regression was chosen to model non-linear relationships more effectively than linear models. Binning Linear Regression was implemented to localize regression within discrete data intervals, addressing non-linear trends within smaller, more homogeneous segments of the data [13], [14]. To take into account uneven density distributions, Weighted Linear Regression was utilized, assigning weights inversely proportional to data density to enhance estimates.

A series of Regularized Convolutional Neural Networks (CNNs) were developed to interpret the complex spatial patterns, suited to the sequential format of the inputs. These models can better handle the variability inherent in the dataset, which includes differences in demographic and physiological factors, reducing the potential bias. The networks varied in the number of 1D CNN layers (ranging from 2 to 5), filter sizes

final feature set is optimized for predictive accuracy, enhancing the model's generalizability and reducing overfitting risks.

Table II illustrates the final selected features based on technical analysis and scientific background knowledge. The first seven features, referred to as Group 1, were selected based on technical and statistical analysis, including correlation and RMSE values. The second group initially comprised six features: $T\_Distance$, $T\_offset$, $T\_atm$, $Humidity$, $Gender\_Female$, $Gender\_Male$, were not identified as strong candidates by correlation analysis and regression-based feature importance assessments. However, as outlined in Section II-C1, these features are biologically and physiologically significant for measuring oral temperature, as supported by a thorough review of relevant literature.

To ensure the inclusion of these biologically essential features, the model was trained under 64 different conditions, encompassing all possible ($2^6$) subsets of these six features to identify the subset that achieved the lowest validation RMSE. The best combination, excluding the $T\_Distance$ feature, was ultimately selected. Further explanations about these combinations are discussed in Section III. The five features from this group that yielded the lowest RMSE were included in the final model, as indicated in Table II. The criteria for selecting these two feature groups ensure both strong predictive power and physiological relevance, thereby enhancing the model's accuracy and reliability in practical applications.

To enhance our model's capacity to capture complex, nonlinear relationships inherent in our dataset, as discussed in

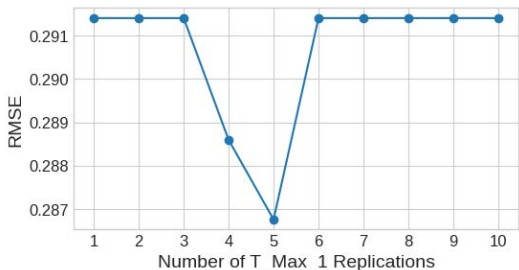

Fig. 2. RMSE Variation with Repetitions of Feature $T\_Max\_1$: Shows the relationship between RMSE and the number of repetitions for $T\_Max\_1$. A minimum RMSE is observed at five repetitions, beyond which additional repetitions do not improve model performance.

TABLE IV
COMPARATIVE ANALYSIS OF DIFFERENT FEATURE SETS FOR PREDICTIVE MODELING, MAE, MSE, AND RMSE.

| Feature Set Description | Number of Features | MAE | MSE | RMSE |
|---|---|---|---|---|
| Correlation-Bio Features Set (a) | 7 | 0.2079 | 0.0746 | 0.2732 |
| Comprehensive Bio Features Set (b) | 13 | 0.2251 | 0.0892 | 0.2986 |
| Optimized Correlation-Bio Set (c) | 12 | 0.2219 | 0.0849 | 0.2913 |
| Expanded Optimal Feature Set (d) | 17 | 0.2134 | 0.0822 | 0.2867 |
| Optimal Combo Feature Set (e) | 15 | 0.1948 | 0.0672 | 0.2592 |
| Final Engineered Feature Set (f) | 20 | 0.1905 | 0.0647 | 0.2545 |

(8 to 64), and kernel sizes (2 and 3), with 'same' padding and strides of 1 to maintain output size.

ReLU activation and L2-regularization were used in the CNN layers to extract key features. A flattening layer was followed by dense layers, starting with 64 units and ReLU activation and ending with a single-neuron output layer for regression. The Adam optimizer was employed to minimize mean squared error. The evaluation of these CNN architectures, summarized in Table VI, helped identify the best configuration for handling Infrared Thermography data.

## III. RESULTS

In this study, we systematically evaluated the performance of various regression models applied to Infrared Thermography data, utilizing three established statistical metrics: Mean Squared Error (MSE), Mean Absolute Error (MAE), and RMSE. These metrics were selected to cater to the diverse nature of our data; MSE highlights the impact of larger errors, MAE offers a straightforward average error magnitude that is less sensitive to outliers, and RMSE provides a balance by giving more weight to larger errors but in a scale comparable to MAE. Collectively, these metrics enable a comprehensive assessment of model accuracy.

### A. Feature Engineering Analysis

The linear regression model without regularization was used as the benchmark to evaluate the effectiveness of the various combinations of features together.

*a) Feature Selection Based on High Correlation and Biological Relevance:* Our analysis began by identifying features that were both highly correlated with the target variable and biologically relevant. The initial selection, named the Correlation-Bio features set in Table IV, included Features

of Table I, which were chosen for their statistical significance and relevance to our study's biological context. This feature set provided a baseline for model performance, achieving an RMSE of 0.2732, MSE of 0.0746, and MAE of 0.2079.

*b) Inclusion of All Biologically Relevant Features:* As the next step, we incorporated all biologically relevant features, regardless of their statistical correlation, in a set named the Comprehensive Bio features set in Table IV. This comprehensive approach was intended to capture the full spectrum of biologically significant factors, expanding the feature set to include Features stated in Section II-C1. This expanded set yielded a slightly increased RMSE of 0.2986, suggesting the inclusion of less predictive features.

*c) Correlation-Based and Comprehensive Biological Features:* In refining our approach, we retained features showing both high correlation with the target variable and biological relevance. For those biologically relevant features with lower correlation, we methodically identified an optimal subset that minimized validation RMSE to benefit from their inclusion in the model performance. This meticulous optimization process led to the development of the Optimized Correlation-Bio set, as denoted in Table IV. This final feature set strikes a balance between statistical robustness and biological insights, achieving an improved RMSE of 0.2913.

*d) Expanding Features Through Repetition of the most correlated feature:* To further enhance model's performance, we experimented with engineering features by augmenting the representation of the most statistically significant feature, Feature $T\_Max\_1$, five-fold. This was based on an analysis of the impact of feature repetition on model performance, as illustrated in our RMSE versus number of repetitions plot as depicted in Figure 2. The plot indicated that repeating this feature five times minimized the RMSE, prompting its inclusion in the final model. This approach led to the formulation of the Expanded Optimal Feature Set, as designated in Table IV, leveraging enhanced feature representation to improve predictive accuracy. The results were improved, with an MAE of 0.2134, MSE of 0.0822, and RMSE of 0.2867.

*e) Using Polynomial combinations of the two most correlated features:* Building upon our refined set of (c), the Optimal Combo Feature Set has polynomial terms of $T\_Max\_1$ and $canthi4Max\_1$ (see Table II) in addition to the Optimized Correlation-Bio set. This set captures complex interactions that linear models might miss, substantially enhancing the model's predictive capabilities. The inclusion of these polynomial features resulted in an improvement in model accuracy, achieving a RMSE of 0.2592, which suggests effective capture of underlying patterns without overfitting.

*f) Final Engineered Feature Set:* The Final Engineered Feature Set, as indicated in Table IV, represents the pinnacle of our feature engineering efforts. It extends the Optimal Combo Feature Set by adding repetitions of the feature $T\_Max\_1$, reinforcing its impact on the predictive model. This integration of repeated and polynomial features aims to maximize the predictive accuracy and robustness of the model. The effectiveness of this strategy is demonstrated by

the lowest recorded MSE of 0.0647 and RMSE of 0.2545 in our experiments, highlighting the benefits of combining multiple feature engineering techniques.

*g) Final Feature Set Validation Using Sequential Backward Selection (SBS):* To validate our final feature set, we employed Sequential Backward Selection (SBS). Starting with the full set of 38 features, including new features from categorical variables mentioned in Section II-B3, SBS iteratively removed the least significant ones, reducing the set to 11. The Ordinary Linear Regression model using the SBS-reduced set achieved an RMSE of 0.3380. SBS was preferred over Sequential Forward Selection (SFS) due to its thorough evaluation of feature combinations and effectiveness in handling irrelevant features. We compared the Ordinary Linear Regression model's performance using the final feature set (Table V) and the SBS-derived set. The RMSE was 0.2545 for the final set and 0.3380 for the SBS set, confirming the final set's superior predictive power and biological relevance.

### B. Regression Models Analysis

We evaluated the performance of various regression models with the final set of selected features using both traditional ML models and deep learning architectures to identify the most effective approach for Infrared Thermography data. Table V compares traditional machine learning models. Table VI analyzes different CNN architectures with varying configurations.

The 1-Nearest Neighbor (1NN) model served as a basic benchmark, and with an RMSE of 0.3873, it demonstrated limited capability in generalizing the dataset's intricate patterns. In contrast, Ordinary Linear Regression and Random Forest provided significantly better results, with an RMSE of 0.2545 and 0.2460, respectively, indicating a strong linear relationship in the data that this model captured effectively.

While KNN, optimized for the number of neighbors, and Support Vector Regression (SVR) offered moderate performance, they did not surpass the linear regression model in terms of overall effectiveness. The RMSE of KNN was 0.2589, and of SVR was 0.2692, suggesting potential misalignments with the data's underlying patterns or insufficient capturing of non-linear interactions by the employed kernel.

Specialized techniques such as Binning emerged as particularly effective, achieving the lowest RMSE of 0.2296 among all tested traditional models. This method's success underscores the utility of segmenting data into bins and applying localized models within these segments to better capture variable dynamics. On the other hand, Piecewise and Weighted Linear Regression underperformed, suggesting issues such as overfitting and suboptimal weighting schemes, respectively.

Quadratic Regression did not substantially improve prediction accuracy, showing that simpler models achieved comparable results without the additional computational complexity.

The use of CNNs necessitated an additional layer of analysis, shown in Table VI. The architecture with four Conv1D(16) layers with a kernel size of 3 and L2-regularization of 0.01 exhibited the best performance among CNNs, with an RMSE of 0.2223, showcasing its capability to handle complex patterns

TABLE V
PERFORMANCE METRICS OF VARIOUS MACHINE LEARNING MODELS USING THE FINAL SET OF SELECTED FEATURES, HIGHLIGHTING MAE, MSE, AND RMSE. THE TABLE INCLUDES A COMPREHENSIVE COMPARISON ACROSS ALL CONSIDERED METHODS.

| Model | Performance measures | | |
|---|---|---|---|
| | MAE | MSE | RMSE |
| 1NN | 0.2748 | 0.1500 | 0.3873 |
| Ordinary Linear Regression | 0.1905 | 0.0647 | 0.2545 |
| KNN with Optimization over K | 0.2005 | 0.0670 | 0.2589 |
| Support Vector Regression | 0.1918 | 0.0725 | 0.2692 |
| Binning | **0.1708** | **0.0527** | **0.2296** |
| Piecewise Linear Regression | 0.2377 | 0.1072 | 0.3273 |
| Weighted Linear Regression | 0.2320 | 0.0925 | 0.3041 |
| Quadratic Regression | 0.2203 | 0.0963 | 0.3103 |
| Random Forest | 0.1907 | 0.0606 | 0.2460 |

TABLE VI
COMPARATIVE ANALYSIS OF DIFFERENT FEATURE SETS FOR PREDICTIVE MODELING, MAE, MSE, AND RMSE.

| Hidden Layer Architecture | Kernel Size | L2-Regularization | Performance Measures | | |
|---|---|---|---|---|---|
| | | | MAE | MSE | RMSE |
| $2 \times Conv1D(64)$ | 2 | 0.01 | 0.2017 | 0.0667 | 0.2583 |
| $2 \times Conv1D(32)$ | 2 | 0.01 | 0.2063 | 0.0989 | 0.2496 |
| $4 \times Conv1D(16)$ | 2 | 0.01 | 0.187 | 0.1496 | 0.2417 |
| $5 \times Conv1D(8)$ | 2 | 0.01 | 0.234 | 0.1552 | 0.2733 |
| $4 \times Conv1D(16)$ | 3 | 0.01 | **0.1823** | **0.1501** | **0.2223** |
| $5 \times Conv1D(16)$ | 3 | 0.01 | 0.1886 | 0.1752 | 0.2298 |
| $4 \times Conv1D(16)$ | 3 | 0.001 | 0.2312 | 0.1208 | 0.3124 |

and non-linearities more effectively. The L2 regularization coefficient was optimized over various values, with 0.01 as the optimal value for minimizing the RMSE.

In summary, the model evaluation revealed that while some advanced models, including certain CNN configurations, enhanced performance, simpler approaches like Binning also provided a highly effective balance of simplicity and accuracy.

## IV. DISCUSSION

The integration of machine learning algorithms with Infrared Thermography (IRT) has demonstrated significant improvements in the accuracy and reliability of non-contact temperature measurements. By employing a comprehensive feature selection strategy that combines both physiological relevance and statistical significance, the models achieved higher predictive accuracy, as evidenced by lower RMSE values compared to more traditional feature-selection methods.

The importance of selecting features based on both scientific literature and empirical data was highlighted. Features such as maximal and inner canthi temperatures were identified as critical predictors due to their strong correlation with core body temperatures. Additionally, the inclusion of environmental factors like humidity and temperature offset further enhanced the model's robustness and reliability in practical applications. Our results outperform those reported in [6], which utilized only a single feature for regression implementation, either TCEmax or Tmax. In [6], the best RMSE values achieved using Ordinary, Piecewise, Weighted, and Binning methods were 0.33, 0.35, 0.30, and 0.45, respectively, when using the optimal choice between TCEmax or Tmax for each method. By contrast, our use of the selected features has resulted in

better alignment and robustness between the predicted outputs of the regression models and the reference temperature values, as demonstrated in Table V. We also introduced CNNs for the thermography dataset as a new layer of analysis that effectively handled complex patterns and non-linearities (see Table VI).

Our model enhances IRT accuracy by addressing discrepancies between measured surface and actual core body temperatures, improving upon traditional methods with an RMSE of 0.2223°C, compared to potential errors up to 2°C in NCITs. NCITs often suffer from sensor inaccuracies and environmental factors that introduce considerable errors. Similarly, unless properly calibrated, IRT systems face reliability challenges. When properly calibrated, IRTs demonstrate enhanced performance over NCITs, achieving a clinical bias within ±0.03°C [6]. Our approach thus offers more reliable temperature measurements and improves the calibration process to ensure high accuracy and robustness in clinical and public health settings. To integrate our approach into a real thermometer device, the IRT system captures thermal images, feeding raw data into our ML model to refine temperature estimates.

The methodologies developed in this study, including the comprehensive feature selection process and the integration of machine learning algorithms with IRT (Figure 1), are not limited to the specific application of infrared thermography for temperature measurement. They can be generalized to other biomedical applications requiring non-contact measurements or predictions of biomedical parameters, where advanced data processing and machine learning are essential [15].

The pipeline, encompassing scientific background knowledge and advanced technical feature engineering, can be applied to other healthcare and sensor applications. For instance, predicting physiological parameters like heart rate, oxygen saturation, or glucose levels using non-contact or remote sensing technologies could benefit from a similar approach [16]. Moreover, this methodology could improve the robustness and accuracy of ill-posed regression problems in non-invasive cancer microstructural imaging methods, offering a substantial improvement over conventional optimization techniques [17]. Comparable machine learning algorithms have been successfully applied in health monitoring devices, enhancing predictive accuracy and reliability [18]. The versatility and potential of these algorithms are further exemplified in diverse applications [19]–[22]. Enhancing predictive accuracy and reliability through robust feature selection and model optimization is widely applicable in biomedical engineering

## V. Conclusion and Outlook

The integration of machine learning algorithms with Infrared Thermography (IRT) has demonstrated significant improvements in the accuracy and reliability of non-contact temperature measurements. By employing a comprehensive feature selection strategy that combines both physiological relevance and statistical significance, the models achieved higher predictive accuracy, as evidenced by lower RMSE values compared to traditional methods. The refined feature set, identified

through a combination of statistical analysis and domain-specific insights, considerably enhanced the model's predictive accuracy while maintaining computational efficiency.

Future research could focus on expanding the applicability of these methods to other infectious diseases and sensor applications, suggesting broad potential for non-contact diagnostic tools and public health advancements.

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
