### B. Preprocessing

In the preprocessing phase of our analysis, we implemented a series of steps to prepare the Infrared Thermography Temperature Dataset for subsequent machine learning tasks. The process involved loading, cleaning, encoding, and transforming the data to ensure a reliable and meaningful dataset for training our models. Here's a description of each step undertaken:

*1) Handling Missing Data:* We initially removed completely empty features, which could indicate unrecorded attributes or experimental artifacts. Additionally, data points with missing values ('NaN's) were eliminated to maintain the dataset's integrity. Both methods resulted in a reduction in the dimensionality of the dataset.

*2) Feature Transformation for Temperature Readings:* Note that each temperature was measured over four rounds. For these temperature readings, the mean was calculated to create a single representative value. This approach reduced the dataset's complexity and noise while retaining the essential thermal profile information.

*3) Standardization of Numerical Features:* Standardization of numerical features is performed to normalize the range of independent variables. Standardization involves subtracting the mean and dividing by the standard deviation to achieve unit variance. This is a critical preprocessing step to ensure unbiased and effective model performance across various machine learning algorithms.

### C. Feature Extraction

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

### E. Model Training and Evaluation

In our study, various regression models were employed to address the challenges presented by Infrared Thermography data, each selected for its unique ability to capture and analyze spatial and temporal patterns.

TABLE II
FINAL ENGINEERED FEATURE SET BASED ON TECHNICAL ANALYSIS AND SCIENTIFIC BACKGROUND KNOWLEDGE

| Final Selected Features | Selection Criteria |
| --- | --- |
| $T\_Max\_1$ | High correlation and low RMSE, strong predictive power (+5X replications) |
| $canthi4Max\_1$ | High correlation and low RMSE, strong predictive power |
| $canthiMax\_1$ | High correlation and low RMSE, strong predictive power |
| $Max1R13\_1$ | Significant correlation and acceptable RMSE, good predictive power |
| $Max1L13\_1$ | Significant correlation and acceptable RMSE, good predictive power |
| $aveAllL13\_1$ | Moderate correlation and acceptable RMSE, contributing to model predictive power |
| $aveAllR13\_1$ | Moderate correlation and acceptable RMSE, contributing to model

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

 Convolutional Neural Networks (CNNs) necessitated an additional layer of analysis, as detailed in Table VI. The architecture containing four Conv1D(16) layers with a kernel size of 3 and L2-regularization of 0.01 exhibited the best performance among CNNs, with an RMSE of 0.2223, showcasing its capability to handle complex patterns and non-linearities more effectively than traditional models. The L2 regularization coefficient was optimized over various values, with 0.01 as the optimal value for minimizing the RMSE.

In summary, the model evaluation revealed that while some advanced models, including certain CNN configurations, enhanced performance, simpler approaches like Binning also provided a highly effective balance of simplicity and accuracy.

## IV. DISCUSSION

The integration of machine learning algorithms with Infrared Thermography (IRT) has demonstrated significant improvements in the accuracy and reliability of non-contact temperature measurements. By employing a comprehensive feature selection strategy that combines both physiological relevance and statistical significance, the models achieved higher predictive accuracy, as evidenced by lower RMSE values compared to more traditional feature-selection methods.

The importance of selecting features based on both scientific literature and empirical data was highlighted. Features such as maximal and inner canthi temperatures were identified as critical predictors due to their strong correlation with core body temperatures. Additionally, the inclusion of environmental factors like humidity and temperature offset further enhanced the model's robustness and reliability in practical applications. Our results outperform those reported in [6], which utilized only a single feature for regression implementation, either TCEmax or Tmax. In [6], the best RMSE values achieved using Ordinary, Piecewise, Weighted, and Binning methods were 0.33, 0.35, 0.30, and 0.45, respectively, when using the optimal choice between TCEmax or Tmax for each method. By contrast, our use of the selected features has resulted in better alignment and robustness between the predicted outputs of the regression models and the reference temperature values, as demonstrated in Table V. Additionally, we introduced CNNs for the thermography dataset, providing a new layer of analysis that effectively handled complex patterns and non-linearities, as shown in Table VI.

The methodologies developed in this study, including the comprehensive feature selection process and the integration of machine learning algorithms with IRT (Figure 1), are not limited to the specific application of infrared thermography for temperature measurement. They can be generalized to other biomedical applications requiring non-contact measurements or predictions of biomedical parameters, where advanced data processing and machine learning are essential [20].

The pipeline, encompassing scientific background knowledge and advanced technical feature engineering, can be applied to other healthcare and sensor applications. For instance, predicting physiological parameters like heart rate, oxygen saturation, or glucose levels using non-contact or remote sensing technologies could benefit from a similar approach [21]. Moreover, this methodology could improve the robustness and accuracy of ill-posed regression problems in non-invasive cancer microstructural imaging methods, offering a substantial improvement over conventional optimization techniques [22]. Comparable machine learning algorithms have been successfully applied in health monitoring devices, enhancing predictive accuracy and reliability [23]. The versatility and potential of these algorithms are further exemplified in diverse applications [24]. The ability to enhance predictive accuracy and reliability through robust feature selection and model optimization is broadly applicable across various domains within biomedical engineering.

## V. CONCLUSION AND OUTLOOK

The integration of machine learning algorithms with Infrared Thermography (IRT) has demonstrated significant improvements in the accuracy and reliability of non-contact temperature measurements. By employing a comprehensive feature selection strategy that combines both physiological relevance and statistical significance, the models achieved higher predictive accuracy, as evidenced by lower RMSE values compared to traditional methods.

The refined feature set, identified through a combination of statistical analysis and domain-specific insights, significantly enhanced the model's predictive accuracy while maintaining computational efficiency.

Future research could focus on expanding the applicability of these methods to other infectious diseases and sensor applications. The adaptability of this approach suggests its potential for broader applications in non-contact diagnostic tools, paving the way for advancements in public health screening and monitoring.