# OpenReview forum: "Fever Detection with Infrared Thermography: Enhancing Accuracy through Machine Learning Techniques"
_IEEE.org/EMBS/BHI/2024/Conference — IEEE BHI'24_

### Official Review · Reviewer_HQca · 2024-08-10
**Accept with minor revisions**

**Overall Rating:** 6
**Confidence:** 3

**Other Quality Metrics:**

Carity of writing: excellent
Clinical significance: excellence
Methodological novelty: fair
Experiments and results: good

**Questions For The Authors:**

Revision suggestions/comments
=============================
Introduction
------------
- In the beginning of the "Introduction" section the phrasing should be improved to be clearer. As it is now, it is mentioned that COVID-19 is a huge crisis and IRT emerged as a technological advancement to support it. This is not true. IRT is a technology which existed before COVID-19.

Methodology
-----------
- [A. Dataset] "The Infrared Thermography Temperature Dataset, sourced from Forward Looking Infrared (FLIR) data for both groups 1 and 2, ..."=> The groups 1 and 2 should be explained. How are they split?
- [B. Preprocessing - Handling missing data] Indeed, eliminating data points with missing values is a commonly used approach. However, there are also other approaches which could be used (e.g. replace missing values after a regression - to mention a very simple approach). I would suggest to explain why the selected approach was applied.
- [D. Feature selection - Technical and statistical analysis] Similarly with my previous comments, I would like to know the reasons that led the authors to focus on the proposed approaches. Even if these are intuitive decisions, I would suggest to discuss why the authors think the selected statistical measures are appropriate compared with alternatives.

Discussion
----------
- As far as I can understand (perhaps I miss somethin) in the current version of the manuscript, an ML model is presented aiming to predict the temperature (oral temperaure) based on IRT data. Perhaps it would make sense to explain a little better how the ML model would be integrated and support a real IRT thermometer device. Would it be used to identify a potential gap between the measured value and the one expected? How would the final measurement be improved?

Editing suggestions/comments
============================
N/A

**Strengths:**

Indeed, IRT is an important diagnostic tool which could be very useful for COVID-19 and beyond. Any method improving its results could have very big impact in every-day clinical practice.

**Summary Of The Paper:**

The paper proposes the use of a Machine Learning (ML) approach to improve the performance of Infrared Thermography (IRT). The subject of the paper is of interest and well-written. My comments should be received only as minor improvement suggestions.

**Weaknesses:**

There are some questions about methodological choices but these should be considered minor issues in my view.

---

### Official Review · Reviewer_pVuF · 2024-08-10
**Evaluation of "Fever Detection with Infrared Thermography: Enhancing Accuracy through Machine Learning Techniques"**

**Overall Rating:** 7
**Confidence:** 4

**Other Quality Metrics:**

According to the above things that was analyzed in this paper the metrics are the following:
Clarity of Writing: Good
Clinical Significance: Fair
Methodological Novelty: Good
Experiments and Results: Good

**Questions For The Authors:**

1. Now one might say, which brings us to our most important question in the context of grants- how do you plan on overcoming the bias inherent within your data set so that methods as proposed can be utilized among all populations?

2. How has the clinical validation of your approach compared to current standard practices?

**Strengths:**

1. Novel Use Case: The combination of machine learning and infrared thermography is a unique method for fever detection that may provide additional diagnostic benefit.

2. Full Assessment: The research conducts a rigorous assessment of various machine learning models and feature selection approaches, thus demonstrating their viability.

**Summary Of The Paper:**

This paper proposes an advanced approach to fever detection using infrared thermography combined with machine learning techniques. The authors aim to enhance the accuracy of fever detection by employing various machine learning models and feature selection strategies, ultimately achieving improved performance compared to traditional methods.

**Weaknesses:**

1. The proposed methods are not sufficiently clinically validated, and thus to be able to know how they might work in real-world application.

2. Database Restrictions: As part of the data was extracted from a small dataset, it could affect to whether or not we can extrapolate our finding.

---

### Decision · Program_Chairs · 2024-09-23

Accept